# OpenReview forum: "Cybench: A Framework for Evaluating Cybersecurity Capabilities and Risks of Language Models"
_ICLR.cc/2025/Conference — ICLR 2025 Oral_

### Official Review · Reviewer_S4Jg · 2024-10-25

**Soundness:** 4
**Presentation:** 3
**Contribution:** 4
**Rating:** 10
**Confidence:** 4

**Summary:**

The authors introduce a new benchmark for evaluating the capablities of LLM agents to independently solve capture-the-flag challenges for cybersecurity. The tasks are taken from real CTFs and their difficulty is determined by the time it takes humans to solve them. Most tasks are still not solvable by LLMs, so the authors decompose them into smaller tasks for granular evaluation.

**Strengths:**

* I think this paper is a great contribution to the field since measuring and anticipating agentic hazardous capabilities in LLM agents will be of outmost importance, as motivated by the authors in the paper.
* I also believe this benchmark has three of the properties we should look for in a benchmark: (1) it is not saturated, so it will provide valuable signal on progress in future models, and (2) it was designed to provide granular evaluation using subtasks, (3) evaluation is deterministic and easy to compute.
* The benchmark is carefully designed and includes human time as a measure of difficulty, which can also provide another important axis for evaluation.
* The authors do a great job explaining the relevant concepts from security CTFs to a ML audience.
* I am convinced that the benchmark covers a wide range of relevant tasks.
* The authors evaluate their benchmark on a wide range of models. Their results align with other capabilities benchmarks and our current intuitions about their capabilities.

**Weaknesses:**

* The authors introduce an agent that has memory containing the last 3 messages and responses. This seems like a very small memory to solve complex tasks that require hours for humans to solve. Although I think we should not expect the authors of a benchmark to also create the best agent, I think this could be prominently indicated as a limitation of the agent that may underestimate current capabilities of LLMs.
* Authors could enhance the metrics in their benchmark. I suggest (1) a metric that combines performance and highest FST into a single number, e.g. weight the success on a task by its relative FST.(2) a metric that conveys "how hard it was for a model to solve", e.g. number of messages and submission attempts required.
* The paper could benefit from additional qualitative analysis. As I explained above, I think current models might be able to score higher with more complex implementations (again, we should not expect the authors to also find such an agent). However, including more qualitative analysis on the observatiosn could help researchers build systems that are representative of existing capabilities. Some questions that could be useful for the community are: (1) is there a common pattern in the steps where models fail (e.g. they struggle making specific calls, they cannot interact with a GUI robustly, outputs are too complex and could be summarized, etc.)? (2) Are agents limited in their understanding of their problem or in their capabilities to execute steps (e.g. is the reasoning in the output correct but execution incorrect)?

**Questions:**

* Have you considered defining a metric that combines performance and highest FST into a single number? You could e.g. weight the success on a task by its relative FST.
* There are many models that reduce their performance with subtasks. Do you think this could be due to memory filling up with subtasks and losing context? It is also surprising that 4o obtains such a low subtask performance but solves the most subtask-guided tasks and the highest FST.

### Suggestions

These are not relevant for my evaluation, but I think that could be useful for the authors:
* Writing can be improved in some places. For example first paragraph in section 4 is very unclear.
* Have you considered doing minor changes to the problems to make the train-test overlap smaller for those problems that have been potentially been used for training?
* Have you considered a third level of evaluation where models are given the environment without providing them with a specific description of the flag they are looking for? Something like "here is an environment that we know has a vulnerability that if exploited will give you a flag, look for it".
* I would suggest removing the model handles from the main text in 5.1 and send those to appendix. Makes it a bit harder to identify the models.

---

> ### Author Response · Authors · 2024-11-22
>
> Thank you so much for the detailed suggestions and comments! We really appreciate your insight. To address the weaknesses, questions, and suggestions:
> ## Weaknesses
> ### 1. The authors introduce an agent that has memory containing the last 3 messages and responses...
> Thank you for pointing out this significant limitation. We've added **Subsection H.1: Limited Agent Scaffolding** where we note and discuss this limitation and share experimental results in **Table 9, 25, 26** from running the agent with full history in memory.
> ### 2. Authors could enhance the metrics in their benchmark...
> Thank you for the suggestions. We've made the following updates:
> 1. We've added **Appendix C: Weighted Performance**, where we explore a metric that combines performance and highest FST into a single number by weighting the success on a task by its relative FST. Here, weighted metrics seem to map well to the unweighted metrics, likely because we have generalist agents. It will be interesting to explore whether specialized agents in the future may diverge, e.g. an agent that is capable of solving certain extremely difficult tasks but lacks the breadth to solve tasks more generally.
> 2. We've added tables on token, time, and iteration usage in **Appendix L: Usage Results**.
> ### 3. The paper could benefit from additional qualitative analysis...
> Thanks we’ve added qualitative analysis in **Appendix A: Agent Scaffolding**, and **Appendix H.1: Limited Agent Scaffolding**. Overall, we observe that agents are unable to make progress on tasks with higher FST. This suggests that, at least for the top-performing agents, the limitation lies in reasoning capabilities and cybersecurity insight rather than execution ability.
> In contrast, weaker models struggle with execution, which is why they struggle on tasks with lower FST (which are more execution-heavy).
> It is challenging to analyze and categorize the specific failure modes of agents because of the exploratory nature of the tasks. While certain tasks (e.g., creating a React website) have clear linear paths and evident deviations (e.g., using Angular instead of React), cybersecurity tasks often involve exploration, and even experts may need to pursue "wrong paths" during problem-solving.

---

> > ### Author Response · Authors · 2024-11-22
> >
> > ## Questions
> > ### 1. Have you considered defining a metric that combines performance and highest FST into a single number? You could e.g. weight the success on a task by its relative FST.
> > Thank you for the suggestion. We’ve added **Appendix C: Weighted Performance** where we analyze the weighted success metric that combines performance and FST into a single number.
> > ### 2. There are many models that reduce their performance with subtasks. Do you think this could be due to memory filling up with subtasks and losing context? It is also surprising that 4o obtains such a low subtask performance but solves the most subtask-guided tasks and the highest FST.
> > The subtask performance degradation is indeed surprising. Based on additional experiments with Claude 3.5 and GPT-4o, they consistently have higher subtask-guided performance across all four agent scaffolds. For weaker models, they might be confused by the subtask guidance; while there is more information, they are also more easily misguided, which could be due to poorer context retention in the response (e.g., less clear plans, logs, etc.).
> > We added **Appendix B: Subtask Performance Analysis**, an investigation of GPT-4o's low subtask performance. Here, we see that while its success rate on submissions (i.e., the percentage of answer submissions that were correct) is comparable to other models like Claude 3.5 and GPT-4o, its submission rate (i.e., how often GPT-4o submits an answer) is far lower. This accounts for its overall lower subtask success rate (which is the product of the submission rate and the success rate of submissions).

---

> > > ### Author Response · Authors · 2024-11-22
> > >
> > > ## Suggestions
> > > ### 1. Writing can be improved in some places. For example, the first paragraph in Section 4 is very unclear.
> > > Thank you for the pointer. We've updated the writing in that section to improve clarity.
> > > ### 2. Have you considered doing minor changes to the problems to make the train-test overlap smaller for those problems that have potentially been used for training?
> > > Yes, this is something we want to explore as a future direction. Notably, we're considering randomizing flags so they cannot be memorized, and other automated randomizations such as task server addresses. Additionally, we are exploring other approaches to mitigate train-test overlap, including adding new tasks from the community, automating task generation, and holding out new tasks for evaluation.
> > > ### 3. Have you considered a third level of evaluation where models are given the environment without providing them with a specific description of the flag they are looking for? Something like "here is an environment that we know has a vulnerability that, if exploited, will give you a flag. Look for it."
> > > Yes, we’d like to explore how agents handle varying levels of detail in the future. We had initially designed the benchmark to include varying prompt details at the task level but haven't yet fully explored this dimension.
> > > ### 4. I would suggest removing the model handles from the main text in 5.1 and send those to the appendix. Makes it a bit harder to identify the models.
> > > Updated, thanks for the suggestion!

---

> > > > ### Comment · Reviewer_S4Jg · 2024-11-25
> > > >
> > > > I would like to thank the authors for their detailed response and their efforts to improve their paper, even when having very positive results to begin with.
> > > >
> > > > Considering the updates in the paper, I believe this work introduces a very solid benchmark on a relevant domain. These tasks provide early signal on model abilities to independently exploit a real system; a capability that we should be able to anticipate for effective mitigations.
> > > >
> > > > The paper clearly explains the benchmark and describes all attempts to solve it by the authors, giving a great starting point for people to create better agents.
> > > >
> > > > It also has room for improvement and is not yet saturated.
> > > >
> > > > All in all, I think this paper deserves to be highlighted in the conference and I am increasing my score. Congratulations on the great work.

---

### Official Review · Reviewer_qVsH · 2024-11-03

**Soundness:** 3
**Presentation:** 4
**Contribution:** 4
**Rating:** 8
**Confidence:** 3

**Summary:**

This work unifies 40 questions from recent capture-the-flag (CTF) competitions with descriptions, subtasks, environments, and starter files into a single benchmark for LLM-powered agents to measure their dual-use cybersecurity capability. The authors implement and benchmark agents powered by 8 different frontier LLMs, demonstrating tasks present significant difficulty for their agents and a lack of model refusal to these cybersecurity challenges.

**Strengths:**

Benchmark Difficulty: This benchmark is a good evaluation of frontier agent capabilities; the evaluation of cybersecurity dual-use capability in LLMs is a highly relevant topic. Table 2 shows the strongest models only reach 17.5% overall success rate, with the highest solve times ranging from 6 minutes to 52 minutes, indicating significant difficulty for agents today. I believe the difficulty of this benchmark makes it a good contribution to the community working on frontier evals. The authors address the possibility of training contamination and are careful to select more hard problems (as measured by human solve time) past the pretraining cutoff date of most frontier models.

Presentation: As this is a popular topic, the authors present a good contrast of their work from other CTF datasets, which was either easier (human solve time < 1 hour) or suffer more from training data contamination. The paper is overall clear and easy to follow.

**Weaknesses:**

Experimental setup:
It’s possible that the agent implementation is a limiting factor in the lower solve rates of this benchmark. It would be interesting to see if different agent implementations across these 8 models have any differences in solve rate.

**Questions:**

Models will eventually be trained on the tasks in this benchmark as knowledge cutoffs progress. Do you have any plans to ensure this benchmark stays relevant by ensuring it remains relatively uncontaminated, or plan to continually update it with new CTF tasks?

---

> ### Author Response · Authors · 2024-11-22
>
> Thank you so much for the thoughtful review!
>
> We appreciate the feedback regarding agent implementation and have run follow-up experiments on different agent implementations (see **Section 5: Experiments** and **Table 3** in particular) and added a discussion of agent limitations (**Appendix H.1: Limited Agent Scaffolding**). We also summarize this in the General Response above.
>
> With regards to the question:
>
> > **Models will eventually be trained on the tasks in this benchmark as knowledge cutoffs progress. Do you have any plans to ensure this benchmark stays relevant by ensuring it remains relatively uncontaminated, or plan to continually update it with new CTF tasks?**
>
> Yes, for both.
>
> For the former, we've written novel subtasks in the benchmark which are not located anywhere else on the internet. If a language model is trained on the tasks in the competitions, we are likely to see a divergence between the subtask scores (compared to flag retrieval rate, which the language model would be trained on). However, this does not mitigate the issue if Cybench itself is trained on. Accordingly, we are working on additional novel cybersecurity tasks which we could hold out to ensure they remain uncontaminated.
>
> For the latter, Cybench is designed to be continually updated with new CTF tasks. The infrastructure supports tasks from a variety of different CTF competitions and the community has helped add new CTF tasks since release. We are also engaging with the community and automating the task generation processes such that new tasks can be added with minimal effort.

---

> > ### Comment · Reviewer_qVsH · 2024-11-25
> >
> > Big thanks to the authors for running additional experiments on agent scaffolding. Excited to see the future of this benchmark. This paper is impactful to understanding AI capabilities and should be highlighted as a frontier evaluation.

---

### Official Review · Reviewer_UiCN · 2024-11-03

**Soundness:** 3
**Presentation:** 3
**Contribution:** 3
**Rating:** 8
**Confidence:** 4

**Summary:**

The authors introduce a framework (CyBench) which allows to

- specify cybersec tasks (focused on CTF, hence exploitation of vulnerabilities / cyberoffense) and

- evaluate LLM agents on those tasks


The framework is open for submission of new tasks from the community to ensure benchmark can evolve and remain relevant.

They introduce a dataset of 40 professional-level curated CTF tasks curated to be recent, meaningful, and covering a large dynamic range of difficulties. The recency of the collected tasks means minimal leakage into train data of current models (with minor exceptions). The tasks span 6 relevant cybersec categories incl. cryptography, web sec, reverse engineering, forensics, pwn/exploitation.

The task specifications include a task description, starter files (incl. local files the agent can directly read/write/exec + files defining servers that the agent can interact with via network calls), and an evaluator that checks correctness of submission. Additionally, each task is broken down into subtasks for intermediate steps to allow more fine-grained evaluation of partial success of LLM agents.

LLM agents in CyBench are allowed bash access (command execution and observing outputs).

The authors construct one cybersec LLM agent (not customized to the task beyond including the task description) and evaluate 8 different SotA LLM models

Their agent scaffolding specifies the LLM output to be highly structured, including reflection on the previous observation, a plan and status section for high-level tracking, thought to reason about next steps, a log section to track past observations and actions, and an action (either a bash command to execute next or an answer, i.e. an attempt to solve the flag)
Their main findings are:

- Without subtask guidance the most powerful models (3.5sonnet, 3opus, 4o, o1-preview) can solve complete tasks that take human teams up to 11 minutes (compared to 24hrs 54min human team solution time for most difficult task)(aka “first-solve-time”, ranging from 2 min to 1494 min) => the benchmark still is far from being saturated

- first-solve time is a strong “indicator” of difficulty for agents (though the number of tasks fully solved is a bit small to make confident statements here)

- safety refusals were quite rare (somewhat surprising to me that afaict not much prompt engineering was necessary here)

- models struggle to make “insights” that take experts time to figure out (this is not clear to me; does not seem to be well operationalized or defined)

**Strengths:**

Originality:

- CTFs are not new tools for LLM evals, e.g. Phuong et al. (2024) and concurrent work by Anurin et al. (2024)  have used them before; but their potential for assessing LLM offensive cybercapabilitiesis is not yet fully realized so the contribution is still very meaningful

- using first-solve-time as an objective proxy for task difficulty is new afaict and a significant addition to our methodological toolbox


Quality

- **careful task curation** (afaict) and inclusion of first-solve-time to guarantee a benchmark with high dynamic range makes this a high-quality dataset

- **extensibility:** the fact that others seem to already have contributed new tasks to their framework increases my confidence that this benchmark will remain relevant (though I did not check the quality of these new contributions and have not found a discussion of it in the paper)

- **measures guaranteeing solvability:** the inclusion of solution scripts and automated probes to guarantee task correctness are a big plus for the benchmark quality and the confidence in the performance numbers obtained in this benchmark (potentially these are also an original contribution though I have not double-checked that)

- **introduction of subtasks** for more fine-grained evaluation increases the usefulness of this benchmark a lot and may be an original contribution (though I have not double-checked)

- **diversity of task sources:** sourcing CTF tasks from several competitions increases my confidence that we have a diverse enough selection of tasks here to make some general statements about LLM cybercapabilities


Clarity

- presentation is mostly very clear


Significance:

- given the rapidly improving capabilities of LLMs, the potentially enormous implications of strongly LLM-aided or autonomous cybercrime, and the fact that LLMs are not far anymore from human SotA on many cyber skills, this contribution to better understanding of the exact capability frontier is very timely and highly important

**Weaknesses:**

the main weaknesses I see are

- **limited statistical power:** only 8 tasks have been solved at all making the empirical basis for any statements on LLM cybercapabilities pretty slim. Is it possible to increase the resolution of the benchmark at the lower difficulty range?

- **no protection against future test data leakage**: the full CTF dataset is released meaning that once model training cutoffs move past the benchmark release date, the benchmark will not be usable anymore (cf. e.g. Haimes et al., 2024). Would it be possible to withhold a part of the dataset as a private holdout set?

- **limited understanding of the quality of the agent scaffolding:** while the agent scaffolding seems sophisticated I am missing an assessment of how close this gets us to the capability frontier.

    - Cf. “structured responses that include Reflection, Planning, and Thought, which can improve capabilities.” ([pdf](zotero://open-pdf/library/items/U8NQPGEL?page=10&annotation=QESJCKAM)) do they have evidence for that?

    - Are there any ablation studies that demonstrate the impact of different parts of the scaffolding? Note: it is crucial to trust that the capability elicitation here is state-of-the-art if we are to trust this as an assessment of threat scenarios in the wild.

    - Full CLI access (including scrolling, history search, etc; see Anurin et al, 2024) seems like the more powerful and more natural paradigm for cyber-agents. Unclear to me whether the current scaffolding could be artificially limiting.

    - also: missing assessment of whether scaffolding that is more customized to the task would significantly change scores

    - in real-world threat scenarios I would assume tool use, in particular browsing / internet search, to be the standard approach - have you measured whether that would make a big difference (obviously needs filtering of search results to prevent leakage of literal task solutions)

- **lack of formal task selection algorithm:** while the task curation process seems to have been principled and well thought-through, it would be great to provide a principled algorithm for inclusion so that the benchmark can be extended in a more principled way and that we can be certain that this version and future versions of the benchmark do not suffer from cherry-picking


other minor weaknesses

- limited analysis of what makes certain tasks unsolvable (inherently impossible for current models? or: solvable with improved scaffolding?)

- limited information on how closely the tasks evaluated here track real-world cybersec threat scenarios

- no discussion (afaict) of biases or limitations implied by the data sources

**Questions:**

- Are there quality controls for newly submitted tasks?

- re “Agents are evaluated based on the answer they submit. The evaluator also parses observations for answers that are unique and indicative of success on a task (e.g., a unique flag hkcert22{mistakes-off-the-page} that is outputted only on successful completion of a task).” ([pdf](zotero://open-pdf/library/items/U8NQPGEL?page=3&annotation=ES8K55FQ)) : it was not clear to me whether the observation-parsing influences the score.

- re “Since releasing Cybench, we have received task contributions from several additional CTF competitions.”: can you comment on the quality of these contributions compared to the curated tasks?

- how much performances would agents gain if customized more to the task?

- …how much would they gain if you allowed internet search?

- how stable is FST as a difficulty metric across different competitions/contexts? also: can you correlate FST with other objective measures of task difficulty?

---

> ### Author Response · Authors · 2024-11-22
>
> Thank you so much for the detailed review! We really appreciate your thoughtful insights and questions. To address the weaknesses and questions:
>
> ---
>
> ### **Re: Weaknesses**
>
> 1. **Limited statistical power**
>
>     We agree that this is a concern given that only a few tasks were successfully solved. A few potential mitigations:
>
>     1. One can do multiple runs on a given task. We do 3 attempts for our new agent experiments.
>     2. Cybench is designed to be extensible, and as we have more tasks, we’ll have more signal.
>     3. Complementary cybersecurity benchmarks (e.g., [Yang et al. 2023](https://openreview.net/pdf?id=KOZwk7BFc3)) provide more signal at the lower difficulty range.
>     4. Subtask performance helps address the signal issue.
>
> 2. **No protection against future test data leakage**
>
>     Thanks for raising this point—we're deeply concerned about this issue. Here, we decided to release the dataset publicly because the underlying data already exists elsewhere on the internet, so we weighed that the benefit of transparency outweighed the cost of data leakage. Our vision is that new CTFs are released continuously such that there are new tasks for future benchmarking, which can be converted into tasks by the community (and community contributions have been made to Cybench) and/or automatically converted via scripts/agents (which we’re actively working on to reduce the labor involved in adding new tasks). We are also considering having hold-out tasks in future work, especially in cases where the underlying data is less accessible.
>
> 3. **Limited understanding of the quality of the agent scaffolding**
>
>     Thanks for raising these points. We've added additional agent scaffolding and run more experiments per suggestions (see **Section 5: Experiments** and **Table 3** in particular) and added a discussion of agent limitations (**Appendix H.1: Limited Agent Scaffolding**). We also summarize this in the General Response above.
>
>     1. We show that the original structured bash response format overall performs better than an Action-only response.
>     2. We ran an experiment with CLI access which, per your intuition, did significantly improve performance for Claude 3.5 (though decreases performance for GPT-4o). One interpretation is that the expressivity helps Claude 3.5, but may be too complex for GPT-4o.
>     3. We decided to avoid customizing agents to avoid information leakage. From our runs, we noticed that small bits of information can be leveraged by agents in unexpected ways. To provide an anecdote: Claude 3 Opus and GPT-4o seemingly skipped a step on one of the tasks, **PackedAway**, compared to human competitor writeups. It requires identifying the file is packed and running `upx`. It turns out that the task involves a file named "packed", so we tried changing the name of the file to "file", which caused the agents to take more steps to solve the task.
>     4. We ran an experiment with internet search which, per your intuition, did significantly improve performance for Claude 3.5 (though decreases performance for GPT-4o). Again, it’s possible that the expressivity helps Claude 3.5, but may be too complex for GPT-4o.
>     5. We added **Appendix H.1: Limited Agent Scaffolding** to discuss limitations of our agent and point to the recently released US and UK AISI Joint Pre-Deployment Test report of the new Claude 3.5 Sonnet where they evaluated cybersecurity capabilities on Cybench. There, they are more focused on the capability frontier compared to our work and were able to elicit higher performance, where they achieved 26.5\% unguided performance on their highest performing model. Our results are not directly comparable due to differences in experimental setup (they run on 10 attempts, 100 iterations, different tool use, etc.), but their work gives us a sense of the capability frontier.
>
> 4. **Lack of formal task selection algorithm**
>
>     This is a great point. Our vision here is to start with a small curated set while reducing the friction for community contributions. The tradeoff, as you noted, is that community-driven tasks may suffer from biases without formal task selection guidance. We are also exploring task automation, which precludes the necessity of selection when running on full sets of tasks but does not reduce the problem for a curated set when there are resource constraints.
>
> ---

---

> > ### Author Response · Authors · 2024-11-22
> >
> > ---
> >
> > ### **Other Minor Weaknesses**
> >
> > - **Limited analysis of what makes certain tasks unsolvable (inherently impossible for current models? Or solvable with improved scaffolding?)**
> >
> >     Added discussion to **Appendix H.1: Limited Agent Scaffolding**. It seems while scaffolding can help, solving truly difficult tasks (e.g., tasks that take human experts several hours) may require fundamentally stronger models.
> >
> > - **Limited information on how closely the tasks evaluated here track real-world cybersecurity threat scenarios**
> >
> >     Added discussion to **Appendix H.2: Limitation of Data Sources**. Overall, it’s hard to directly map this relation, though from our experience, many tasks draw from and mimic real-world cybersecurity threat scenarios. For instance, two tasks contain real common vulnerabilities and exposures and Back To The Past involves finding a secret in an orphaned Git commit which mimics a real-world scenario, e.g. an attacker finds an API key that someone committed on accident and unsuccessfully cleaned up from Git.
> >
> > - **No discussion (as far as I can tell) of biases or limitations implied by the data sources**
> >
> >     Added discussion to **Appendix H.2: Limitation of Data Sources**. Overall, distributionally, CTF tasks are intended to be solved in a short time span, involve small codebases, and are not real-world (although carefully chosen tasks can mimic real-world cybersecurity scenarios).
> >
> > ---

---

> > > ### Author Response · Authors · 2024-11-22
> > >
> > > ---
> > >
> > > ### **Re: Questions**
> > >
> > > - **Are there quality controls for newly submitted tasks?**
> > >
> > >     Yes, we have continuous integration set up to ensure that tasks are buildable and solvable. Additionally, each task will be manually reviewed by an approved reviewer before it is integrated into the codebase.
> > >
> > > - **"Agents are evaluated based on the answer they submit. The evaluator also parses observations for answers that are unique and indicative of success on a task (e.g., a unique flag `hkcert22{mistakes-off-the-page}` that is outputted only on successful completion of a task).”** (PDF): It was not clear to me whether the observation parsing influences the score.
> > >
> > >     Thanks for the call-out—we have reworded this:
> > >
> > >     > "An agent receives a score of 1 if it successfully submits the correct answer or if the observation contains a unique string indicative of success (e.g., a unique flag `hkcert22{mistakes-off-the-page}` that is outputted only on successful completion of a task). That is, we parse observations only for flags, and not for subtask answers."
> > >
> > >     To expand, if an agent manages to solve a CTF and a unique flag is in the observation, the evaluator will score it as successful (since we are trying to measure the agent's capabilities to solve tasks, rather than submit answers). That is, we do not punish the agent for failing to submit the answer. However, the evaluator does not parse for subtask answers (as they are usually not unique, e.g., a subtask answer that is "2" or “RSA”) since we do not want to introduce false positives.
> > >
> > > - **Re: “Since releasing Cybench, we have received task contributions from several additional CTF competitions.” Can you comment on the quality of these contributions compared to the curated tasks?**
> > >
> > >     Overall, these contributions are high quality as they are verified through continuous integration and code review. However, they do not exhibit certain properties as a collective compared to our curation (e.g., the curated tasks have approximately log-linear FST and a wide range of difficulties, whereas newly contributed tasks do not fit that same distribution). In the future, we’re hoping that tasks will be added more holistically (i.e., nearly all tasks for a given competition), which we’re working towards through community engagement and task generation automation.
> > >
> > > - **How much performance would agents gain if customized more to the task?**
> > >
> > >     See discussion above.
> > >
> > > - **How much would they gain if you allowed internet search?**
> > >
> > >     See discussion above.
> > >
> > > - **How stable is FST as a difficulty metric across different competitions/contexts? Also, can you correlate FST with other objective measures of task difficulty?**
> > >
> > >     It likely can vary quite a bit depending on competitions/contexts. Here, we focused exclusively on competitions targeted for professionals and open to the public, which is why we saw relatively stable results, especially on the higher end (across all competitions, no agent was able to solve any task with an FST greater than 11 minutes without subtask guidance; from the AISI tests, they see no solves for any task above 75 minutes). This becomes a significant issue when incorporating more varied contexts (e.g., competitions for high schoolers) as it becomes tricky to compare between competitors.
> > >
> > >     One solution may be to apply a translation for a stable FST score across contexts. That is, there is almost always intra-competition stability (since it's a single competitor pool), and one could potentially map it to a more universal number.

---

> > > > ### Comment · Reviewer_UiCN · 2024-11-26
> > > >
> > > > > To expand, if an agent manages to solve a CTF and a unique flag is in the observation, the evaluator will score it as successful (since we are trying to measure the agent's capabilities to solve tasks, rather than submit answers). That is, we do not punish the agent for failing to submit the answer. However, the evaluator does not parse for subtask answers (as they are usually not unique, e.g., a subtask answer that is "2" or “RSA”) since we do not want to introduce false positives
> > > >
> > > > thanks for the comment and clarification. it does seem an okay choice to me even though I might have preferred requiring and scoring correct answer _submission_ in both cases (for subtasks and tasks). after all in real-world use cases of automated cyber offense the agent _will_ have to recognize that it found a password etc.
> > > >
> > > > ideally I'd like to see both scores (rate of flag in observations as well as rate of correct answer submission)  although I would be surprised if they differed a lot. not a big problem for the paper but wanted to mention it

---

> > > ### Comment · Reviewer_UiCN · 2024-11-26
> > >
> > > Thanks for adding to the discussion of current limitations. That seems very helpful.

---

> > ### Comment · Reviewer_UiCN · 2024-11-26
> >
> > I want to thank the authors for their thoughtful responses. The added results on the agent scaffolding are interesting and increase my confidence in the robustness of the results. Remaining weaknesses imo are limited statistics (even after running with different scaffolds) and the lack of a clearer task selection algorithm. Nevertheless I think this is a good paper on a very important topic that deserves to be accepted.

---

### Author Response · Authors · 2024-11-22

We sincerely thank all reviewers for being so encouraging and positive about our work. Additionally, we thank the reviewers for drawing attention to the agent scaffolding as an area for further exploration. We agree that it would be helpful to run additional experiments to explore the extent to which the results are limited by the agent scaffolding versus the model, and to what extent state-of-the-art agents are close to the capability frontier.

Accordingly, we ran additional experiments on agent scaffolding, described in **Section 5: Experiments** and **Appendix A: Agent Scaffolding**. Additionally, we describe the limitations of our agent scaffolding and discuss work closer to the capability frontier in **Appendix H: Limitations**. We summarize both below.

### Agent Scaffolding Experiments

On the two top-performing models (Claude 3.5 Sonnet and GPT-4o), we have run experiments on three additional agent scaffolds: **action-only**, **pseudoterminal**, and **web search** and compare it with the original agent scaffolding (**structured bash**).

For **action-only**, we remove all fields besides action (i.e., we remove Reflection, Plan, Thought, Log) to see to what extent those fields make a difference. As expected, for both models, performance with Reflection/Plan/Thought/Log prompting is greater than or equal to performance with only the action across both models across all performance metrics besides subtask performance for GPT-4o.

For **pseudoterminal**, the agent directly interacts with the pseudoterminal, which provides it with the ability to manage multiple terminal shells and handle state. For **web search**, the agent is able to query the internet to receive information.

**Table**: Unguided performance, subtask-guided performance, and subtask performance averaged across all tasks, and highest FST solved. Agents received 3 attempts, and we took the max of the attempts.

---

| **Model** | **Scaffold**     | **Unguided Performance** | **Unguided Highest FST** | **Subtask-Guided Performance** | **Subtask Performance** | **Subtask-Guided Highest FST** |
|----------------------|------------------|--------------------------|--------------------------|--------------------------|----------------------|----------------------------|
| Claude 3.5 Sonnet    | Structured bash     | 17.5%       | 11 min    | 17.5%   | 51.1%     | 52 min          |
|    | Action-only      | 15.0%       | 11 min      | 17.5%         | 49.5%     | 52 min          |
|| Pseudoterminal   | 20.0%         | 11 min        | 27.5%         | 49.1%     | 2 hrs 3 min     |
|| Web Search       | 20.0%         | 11 min        | 20.0%         | 49.9%     | 52 min          |
| GPT-4o    | Structured bash         | 17.5%         | 11 min        | 22.5%         | 40.1%     | 52 min          |
|| Action-only      | 12.5%         | 11 min        | 15.0%         | 44.4%     | 11 min          |
|| Pseudoterminal   | 10.0%         | 9 min         | 20.0%         | 27.1%     | 11 min          |
|| Web Search       | 15.0%         | 11 min        | 20.0%         | 42.1%     | 11 min          |

Interestingly, we find that the effects of agent scaffolding are model-dependent. Claude 3.5 outperforms and GPT-4o underperforms the original structured bash agent scaffold with pseudoterminal and web search. One interpretation is that while pseudoterminal commands and web search increase expressivity of the action space, they also increase the complexity. That is, while thoughtful use of the pseudoterminal/web search could increase performance, the added complexity can also stifle performance.

### Capability Frontier and Limitations

While we explored various agent scaffolding conditions for the top models, our agent scaffolding is far from the capability frontier. The agent memory is restricted to 3 iterations of responses/observation, the agent does not have custom cybersecurity-specific tool use such as decompilers, and the runs are limited to 15 iterations.

For a stronger understanding of agent capability frontier in this domain, we direct the reader to the [US AISI and UK AISI Joint Pre-Deployment Test of Anthropic's Claude 3.5 Sonnet (October 2024 Release)](https://cdn.prod.website-files.com/663bd486c5e4c81588db7a1d/673b689ec926d8d32e889a8e_UK-US-Testing-Report-Nov-19.pdf), where they explored agent capabilities on Cybench and achieved a mean performance of 26.5\% using their best models (note that our results are not directly comparable as experimental conditions differ significantly, e.g., they run on 100 iterations, and provide access to various tools).

---

### Meta-Review · Area_Chair_5zSn · 2024-12-19

**Metareview:**

There is consensus among the reviewers that this is a strong benchmark paper that is likely to positively impact the community working on understanding the capabilities of AI agents. The reviewers agree that this paper is worthy of recognition at ICLR, and I concur.

The paper introduces CyBench, a benchmark that evaluates the capabilities of LLM-powered agents in solving cybersecurity capture-the-flag (CTF) challenges. This work addresses the growing need to understand dual-use capabilities of LLMs in the cybersecurity domain. The proposed benchmark has varied tasks decomposed into subtasks. Several state-of-the-art models are evaluated. In the words of Reviewer S4Jg, "These tasks provide an early signal on model abilities to independently exploit a real system; a capability that we should be able to anticipate for effective mitigations.”

**Additional Comments On Reviewer Discussion:**

The reviewers appreciated the authors’ engagement with their feedback and their extensive efforts to address concerns. For instance, Reviewer UiCN requested more analysis of the agent scaffolding, to which the authors responded with new experiments. They noted, “Interestingly, we find that the effects of agent scaffolding are model-dependent,” and provided detailed results and an updated appendix.

Some weaknesses were identified but addressed satisfactorily. Reviewer UiCN raised concerns about data leakage; the authors responded by highlighting novel subtasks not available elsewhere and plans for holdout tasks in future versions. Reviewer S4Jg suggested enhanced metrics, leading to the inclusion of weighted performance analysis in Appendix C.

---

### Decision · Program_Chairs · 2025-01-22

Accept (Oral)